# Phytase dose-dependent response of kidney inositol phosphate levels in poultry

**Colleen Sprigg**[1], **Hayley Whitfield**[1], **Emily Burton**[2], **Dawn Scholey**[2], **Michael R. Bedford**[3], **Charles A. Brearley**[1]*

1 School of Biological Sciences, University of East Anglia, Norwich, United Kingdom, 2 School of Animal, Rural and Environmental Sciences, Nottingham Trent University, Southwell, United Kingdom, 3 AB Vista, Marlborough, Wiltshire, United Kingdom

* c.brearley@uea.ac.uk

**Data Availability Statement:** All relevant data are within the paper and its Supporting Information files. Any question pertaining to the data underlying the results presented in the study can be

## Abstract

Phytases, enzymes that degrade phytate present in feedstuffs, are widely added to the diets of monogastric animals. Many studies have correlated phytase addition with improved animal productivity and a subset of these have sought to correlate animal performance with phytase-mediated generation of inositol phosphates in different parts of the gastro-intestinal tract or with release of inositol or of phosphate, the absorbable products of phytate degradation. Remarkably, the effect of dietary phytase on tissue inositol phosphates has not been studied. The objective of this study was to determine effect of phytase supplementation on liver and kidney *myo*-inositol and *myo*-inositol phosphates in broiler chickens. For this, methods were developed to measure inositol phosphates in chicken tissues. The study comprised wheat/soy-based diets containing one of three levels of phytase (0, 500 and 6,000 FTU/kg of modified *E. coli* 6-phytase). Diets were provided to broilers for 21 D and on day 21 digesta were collected from the gizzard and ileum. Liver and kidney tissue were harvested. *Myo*-inositol and inositol phosphates were measured in diet, digesta, liver and kidney. Gizzard and ileal content inositol was increased progressively, and total inositol phosphates reduced progressively, by phytase supplementation. The predominant higher inositol phosphates detected in tissues, D-and/or L-Ins(3,4,5,6)$P_4$ and Ins(1,3,4,5,6)$P_5$, differed from those (D-and/or L-Ins(1,2,3,4)$P_4$, D-and/or L-Ins(1,2,5,6)$P_4$, Ins(1,2,3,4,6)$P_5$, D-and/or L-Ins(1,2,3,4,5)$P_5$ and D-and/or L-Ins(1,2,4,5,6)$P_5$) generated from phytate (Ins$P_6$) degradation by *E. coli* 6-phytase or endogenous feed phytase, suggesting tissue inositol phosphates are not the result of direct absorption. Kidney inositol phosphates were reduced progressively by phytase supplementation. These data suggest that tissue inositol phosphate concentrations can be influenced by dietary phytase inclusion rate and that such effects are tissue specific, though the consequences for physiology of such changes have yet to be elucidated.

## Introduction

The majority of dietary phosphate in animal feed is present as phytate, the mixed metal salt of phytic acid [*myo*-inositol 1,2,3,4,5,6-hexakisphosphate, Ins$P_6$], the primary storage form of

addressed to Charles Brearley, University of East Anglia.

**Funding:** This study was funded by award of a BBSRC Norwich Research Park Doctoral Training Studentship (Ref. BB/M011216/1) to C.S. with contribution from AB Vista. https://biodtp. norwichresearchpark.ac.uk/ The funders had no role in study design, data collection and analysis, decision to publish, or preparation of the manuscript.

**Competing interests:** The feeding trials described in this study were commissioned at Nottingham Trent University by AB Vista. AB Vista had no role in conducting the research, generating the data, interpreting the results or writing the manuscript.

phosphate in plant tissues, but a form with reduced availability for digestion by non-ruminant animals [1, 2]. In order to circumvent the need for the addition of inorganic rock phosphate and to reduce costs to the producer and consumer, poultry diets are now often supplemented with microbial phytase [3] with the intention of accessing phytate-bound phosphate as an economical, environmentally sustainable phosphate source.

In addition to providing bioavailable phosphorus from feed, the benefits of the addition of phytases to animal growth performance have been well documented through use of animal feeding trials [4, 5] and include improved feed conversion ratio as well as the reduction in myopathies such as woody breast disorder which reduce meat quality and result in economic losses for farmers [6].

Biochemical measurements, typically of bone or blood parameters, are routinely undertaken alongside bird performance measurements to quantify physiological effects of the addition of phytases to diets [7–10]. Attention is now being focused on tissues and organs, predominantly by targeted gene expression, e.g., on inositol or phosphate transporters [11–14] and signalling pathways [5, 6, 15] or by metabolomics [16–18]. Both approaches have been complemented by Western blot of transporters or signalling components in tissues such as intestinal mucosa, liver and muscle [6, 19, 20]. Remarkably, given the importance of inositol phosphates and phosphatidylinositol phosphates to intracellular signalling, the study of effect of phytase has not extended to measurement of these molecules in tissues, except for blood [20].

For feed and digesta, freeze dried and milled samples are typically extracted using sodium fluoride and EDTA solution, pH 10 [11, 21] and $InsP_{3-6}$ analysed by high performance liquid chromatography (HPLC) with post column complexation with ferric ion and detection by UV at 290 nm. Commonly, inositol is measured by HPLC-pulsed amperometry [9, 11, 22–24] by GC [25, 26] or by enzymatic assay [16, 17, 27]. For tissue samples, however, the methods of extraction and analyses are markedly different. Analysis of avian erythrocyte inositol phosphates by acid gradient HPLC has been reported [28–30], but for other organs analysis has been limited to *myo*-inositol [6, 27, 31]. Others have concluded that inositol phosphates are absent from human plasma [32, 33] and similarly for chicken [20], while inositol phosphates have been measured in mouse tissues [34].

In the present study, work up of an HPLC method for tissues has allowed for previously unobtainable measurements of inositol phosphate levels in poultry in response to dietary phytase dosage, beside digesta measurements. It was hypothesised that the known significant impact of phytase supplementation on digesta inositol and inositol phosphates would also extend to tissue inositol phosphate levels.

Therefore, the objectives of this study were to investigate the effects of supplemented phytase on the appearance of inositol phosphates in the gizzard and lower ileum of broilers and on tissue inositol and inositol phosphate levels. The data presented in this study and the treatments discussed are abstracted from a larger 2x4 factorial design study that included an additional treatment of supplemented inositol and two levels of digestibility index marker, not described here. The statistics described here have been applied to the data set published in this study.

## Materials and methods

### Study diets

This investigation was carried out with 3 treatment diets of 3 levels of phytase (0, 500 and 6000FTU). One basal diet (Table 1) was formulated to contain adequate levels of all nutrients according to the Ross Management Manual 2018. The study was made up of one diet phase–a starter–offered as a mash diet.

**Table 1. Ingredient composition and calculated nutrient concentrations of the basal diet.**

| Ingredient | Starter | Nutrient | Calculated |
|---|---|---|---|
| Wheat | 63.12% | Crude protein (%) | 21.55 |
| Soybean meal[1] | 30.59% | Poultry AME kcal/kg | 2961.14 |
| Soy oil | 2.70% | Calcium (%) | 0.95 |
| Salt | 0.35% | Total phosphate (%) | 0.73 |
| DL Methionine | 0.17% | Available phosphate[3] (%) | 0.45 |
| Lysine HCl | 0.12% | Phytate P (%) | 0.23 |
| Limestone | 0.95% | Crude fat (%) | 4.11 |
| Dicalcium Phosphate | 1.50% | Poultry ME MJ/kg | 12.39 |
| Vitamin premix2 | 0.50% | Poultry NE Kcal/kg | 1952.36 |

[1]48% minimum declared crude protein; sourced from USA.

[2]Vitamin and Mineral Premix content (per kg diet): Manganese 100 mg, Zinc 88 mg, Iron 20 mg, Copper 10 mg, Iodine 1 mg, Magnesium 0.48 mg, Selenium 0.2 mg, Retinol 13.5 mg, Cholecalciferol 3 mg, Tocopherol 25 mg, Menadione 5.0 mg, Thiamine 3 mg, Riboflavin 10.0 mg, Pantothenic acid 15 mg, Pyroxidine 3.0 mg, Niacin 60 mg, Cobalamin 30 µg, Folic acid 1.5 mg, Biotin 125 µg.

[3] Available phosphate (%) does not account for phytate P contribution

The basal diet was divided into 3 equal parts. One part of each lot remained without phytase (Control). The other parts were supplemented with phytase, Quantum Blue, a thermo-tolerant modified *E. coli* 6-phytase (Quantum Blue, EC 3.1.3.26) supplied by AB Vista (Marlborough, UK). The phytase was added at an intended activity of 500 or 6000 FTU/kg of diet, hereafter Phy500 or Phy6000. The concentrations of inositol and inositol phosphates in each diet were measured by HPLC, and the study diets contained similar concentrations of inositol phosphates (InsPs) (Table 2).

## Animals and management

Three levels of phytase (0, Phy500 or Phy6000) were fed in the absence of Ti (TiO$_2$) as an indigestible marker–one aspect of the trial design, to be published separately, was to measure InsP degradation and the effect of dietary Ti inclusion on InsP measurements. The study was performed at Nottingham Trent University (NTU) Poultry Research Unit, School of Animal, Rural and Environmental Sciences, NTU. Institutional and UK national NC3R ARRIVE guidelines for the care, use and reporting of animals in research [35] were followed during the study and all experimental procedures were approved by Nottingham Trent University's animal ethics review committee (internal code ARE202134). Birds underwent routine vaccination for Marek's Disease and Infectious Bronchitis (IBH120) at the commercial hatchery.

In the trial as a whole, 480 male Ross 308 broilers were obtained from a commercial hatchery (PD Hook, Cote, Oxford, UK) and randomly allocated to 48 pens as part of a larger study on day 1 (0.8 x 0.8 m), with 10 birds per pen with solid floors covered with wood shavings. Data presented in this study is abstracted from the larger trial set, relating to 3 diet conditions,

**Table 2. Measured inositol and inositol phosphate levels of treatment diets[a].**

| Diet | Inositol | InsP$_3$ | InsP$_4$ | InsP$_5$ | InsP$_6$ | Σ InsP |
|---|---|---|---|---|---|---|
| Control | 80 | 260 | 430 | 1660 | 18240 | 20590 |
| Phy500 | 100 | 210 | 430 | 1540 | 18520 | 20700 |
| Phy6000 | 60 | 290 | 750 | 1460 | 15980 | 18480 |

[a] Concentrations given as nmol per g dry weight. Control diet 0FTU phytase.

with 6 replicate pens of 10 birds in each per diet, from which two birds were sampled per pen. The dietary treatments were assigned to the pens (6 pens/treatment) in accordance with a randomized block design in the animal house. Diets were fed from d 1 until slaughter at d 21. Light was provided for 23 hours at placement with 30–40 lux intensity, 1 hour dark, and gradually adjusted to achieve 6 hours of dark by d 7, with 30 minutes of dawn/dusk lighting applied either side of dark period. The temperature of the housing unit was set to 30˚C at d 1 and gradually decreased to 21˚C over the rearing period. Air quality measurements of carbon dioxide and ammonia levels were monitored, with ammonia not exceeding 25ppm. Diets and water were offered for *ad libitum* consumption until euthanasia at d 21.

## Sampling and analytical methods

After all 10 birds / pen were live-weighed, two birds/pen were randomly selected on day 21 post hatch and euthanized via cervical dislocation without prior stunning by a trained personnel in accordance to the Welfare of Animals at the Time of Killing (England) Regulations [2015] guidelines for poultry. From each euthanized bird, the gizzard was excised and opened so the contents could be gently scraped into a 100 ml container as a pooled sample from both birds, prior to storage at -20˚C prior to freeze-drying. For ileal digesta collection, digesta from the same two birds was collected by gentle digital pressure into one pot and stored at -20˚C prior to freeze-drying. Once freeze-dried, the samples were finely ground with a pestle and mortar. The ground samples were stored at 4˚C until analysis. Diets, gizzard and ileal digesta were extracted as described in Whitfield et al. [20].

For tissue analysis, from each of the two birds from which digesta was pooled for analysis, brain, kidney, liver and leg/breast muscle samples were excised, taking care to ensure tissue was consistently excised from the same region of organ or muscle for each bird. For each tissue type collected, the samples of both birds were pooled and stored in polythene bags and immediately frozen at -20˚C before shipping to UEA for inositol phosphate and inositol analysis. Samples were stored thereafter at -80˚C. After defrosting, 100 mg slices of tissue were taken for InsP extraction and analysis.

For inositol phosphate analysis, 100 mg (frozen weight) of kidney tissue was homogenised by Ultra-Turrax (IKA T-10 Ultra-Turrax® High-Speed Homogeniser) in 600 μL: 1M $HClO_4$ on ice and transferred to a 1.5 mL tube. Samples were kept on ice for 20 minutes with vortex mixing every 10 minutes and centrifuged at 13,000 x *g* for 10 minutes at 4˚C. The resulting cleared lysate was transferred to a clean 1.5 mL tube, and 20 μL of which was taken and diluted to 1000 μL with 18.2 Megohm.cm water for inositol analysis.

The following extraction method is adapted from Wilson et al. [32]. All steps were carried out at 4˚C for the prevention of acid degradation of inositol phosphates. Prior to extraction, titanium dioxide ($TiO_2$) beads (Titansphere® $TiO_2$ 5 μM, Hichrom) were washed in 1M $HClO_4$. Then, to each cleared lysate, 5 mg of Titansphere® $TiO_2$ beads in 50 μL $HClO_4$ was added. Samples were vortexed briefly and extracted for 30 minutes with mixing on a rotator. Samples were centrifuged at 3500 x *g* for 5 minutes to pellet the $TiO_2$ beads and the $HClO_4$ supernatant discarded.

In order to elute the bound inositol phosphates, the $TiO_2$ beads were resuspended in 200 μL 3% ammonium hydroxide solution (pH 10) vortexed and incubated with rotation for 5 minutes at 4˚C. Samples were centrifuged at 3500 x *g* for 1 minute and supernatant containing the inositol phosphates were transferred to a clean 1.5 mL tube. A further 200 μL elution in fresh 3% ammonium hydroxide was carried out and the supernatants pooled. Samples were vacuum evaporated until dry and resuspended in 100 μL of 18.2MOhm.cm water for further analysis by HPLC or stored at -20˚C prior to downstream analysis.

50 µL samples were analysed by high-performance liquid chromatography and UV detection at 290 nm after post column reaction with ferric ion, on a 250 x 3 mm Thermo Scientific™ Dionex™ CarboPac™ PA200 column (Dionex™) with a corresponding 3 x 50 mm guard column of the same material. The column was eluted at a flow rate of 0.4 mL/min with an increasing gradient of methanesulfonic acid, derived from buffer reservoirs containing (A) water and (B) 0.6M methanesulfonic acid, by mixing according to the following schedule: time (minutes), % B; 0, 0; 25, 100; 38, 100 [36]. $Fe[NO_3]_3$ in 2% $HClO_4$ was used as the post-column reagent [37] added at a flow rate of 0.2 mL/min. The elution order of InsPs was established using acid-hydrolysed $InsP_6$ standards. Concentration of InsPs was established by reference to UV detector response to injection of $InsP_6$ (Merck).

For inositol analysis, samples extracted as above were diluted 50-fold in 18.2MOhm.cm water. Inositol was determined by HPLC pulsed amperometry of 20 µL aliquots after separation by 2-dimensional HPLC on Dionex CarboPac PA1 and MA1 columns [38].

### Statistical analysis

Inositol, inositol phosphates and total inositol phosphates for data sets presented in this report were compared by multiple T tests with correction for multiple comparisons using the Holm-Šídák method using GraphPad Prism, version 7.0e, for Mac OS X (GraphPad Software, La Jolla, CA). The level of significance for all tests was set at $P < 0.05$. Adjusted P values following T tests are presented in the tables for each data set.

## Results

### Phytate ($InsP_6$) hydrolysis in gizzard and ileal digesta

Supplementation of diet with Phy500 and Phy6000 reduced total inositol phosphates significantly ($P = 0.037$ and $P<0.0001$, respectively) in gizzard contents (Table 3), with reductions in $InsP_6$ and total InsPs increasing with increasing phytase dose. Total inositol phosphate levels were reduced from 14852 ± 817 nmol/g dwt (dry weight) in the Control group to 8608 (±1756) nmol/g dwt at Phy500 and to 1029 ± 183 nmol/g dwt at Phy6000. Phytase at 500 FTU/kg reduced $InsP_5$ and $InsP_6$ significantly ($P = 0.001$ and $P = 0.018$, respectively), from 4721 ± 440 nmol/g dwt and 6872 ± 995 nmol/g dwt, respectively, to 1121 ± 419 and

**Table 3. Inositol and inositol phosphate ($InsP_{2-6}$) levels (nmol/g dwt) in gizzard digesta of day 21 broilers[1,2].**

| Diet | Inositol | $InsP_2$ | $InsP_3$ | $InsP_4$ | $InsP_5$ | $InsP_6$ | $\Sigma$ InsP |
|---|---|---|---|---|---|---|---|
| Control | 348±77 | 193±42 | 590±137 | 2473±484 | 4721±440 | 6872±995 | 14852±817 |
| Phy500 | 900±147 | 677±139 | 1777±728 | 3387±1353 | 1121±419 | 1645±905 | 8608±1756 |
| Phy6000 | 2606±326 | 74±22 | 417±165 | 298±81 | 96±61 | 142±57 | 1029±183 |
| | | | | Probabilities | | | |
| Control vs. Phy500 | 0.037 | 0.037 | 0.261 | 0.8539 | 0.001 | 0.018 | 0.037 |
| Control vs. Phy6000 | 0.0002 | 0.065 | 0.439 | 0.003 | <0.0001 | 0.0002 | <0.0001 |
| Phy500 vs. Phy6000 | 0.005 | 0.009 | 0.187 | 0.137 | 0.137 | 0.187 | 0.009 |

Abbreviations: $\Sigma$ InsP, total $InsP_2$ to $InsP_6$; $InsP_6$, inositol hexakisphosphate; $InsP_5$, inositol pentakisphosphate; $InsP_4$, inositol tetrakisphosphate; $InsP_3$, inositol trisphosphate; $InsP_2$, inositol bisphosphate.

[1]The control group was fed with a diet with 0.45% calculated available phosphate. Groups Phy500 and Phy6000 were fed with the control diet supplemented with 500 or 6,000 FTU of phytase per kilogram of feed, respectively.

[2]Data are given as group means ± SEM, n = 6, of 6 replicate pens with samples pooled from 2 broilers per pen per treatment. Statistical analysis was performed by multiple T tests with correction for multiple comparisons using the Holm-Šídák method for inositol, inositol phosphate and $\Sigma$ InsP data.

**Table 4. Inositol and inositol phosphate (InsP$_{2-6}$) levels (nmol/g dwt) in ileal digesta of day 21 broilers[1,2].**

| Diet | Inositol | InsP$_2$ | InsP$_3$ | InsP$_4$ | InsP$_5$ | InsP$_6$ | Σ InsP |
|---|---|---|---|---|---|---|---|
| Control | 1008±297 | 502±131 | 1358±155 | 2613±306 | 5285±519 | 51587±3269 | 61346±3702 |
| Phy500 | 2434±654 | 1349±173 | 3401±548 | 7037±2014 | 6198±682 | 29190±3803 | 40410±3921 |
| Phy6000 | 10870±2233 | 2535±659 | 1613±357 | 3843±1172 | 432±96 | 1747±385 | 10173±2236 |
| | | | | Probabilities | | | |
| Control vs. Phy500 | 0.102 | 0.018 | 0.028 | 0.107 | 0.311 | 0.008 | 0.018 |
| Control vs. Phy6000 | 0.004 | 0.038 | 0.557 | 0.557 | 0.0001 | <0.0001 | <0.0001 |
| Phy500 vs. Phy6000 | 0.018 | 0.212 | 0.076 | 0.212 | <0.0001 | 0.0001 | 0.0002 |

Abbreviations: Σ InsP, total InsP$_2$ to InsP$_6$; InsP$_6$, inositol hexakisphosphate; InsP$_5$, inositol pentakisphosphate; InsP$_4$, inositol tetrakisphosphate; InsP$_3$, inositol trisphosphate; InsP$_2$, inositol bisphosphate.

[1]The control group was fed with a diet with 0.45% calculated available phosphate. Groups Phy500 and Phy6000 were fed with the control diet supplemented with 500 or 6,000 FTU of phytase per kilogram of feed, respectively.

[2]Data are given as group means ± SEM, n = 6, of 6 replicate pens with samples pooled from 2 broilers per pen per treatment. Statistical analysis was performed by multiple T tests with correction for multiple comparisons using the Holm-Šídák method for inositol, inositol phosphate and Σ InsP data.

1645 ± 905 nmol/g dwt. The "super dosed" group at Phy6000 also showed highly significant reductions (both P<0.001) in InsP$_5$ (96 ± 61 nmol/ g dwt) and InsP$_6$ (142 ± 57 nmol/g dwt).

Total inositol phosphate levels were reduced significantly from 61346 ± 3702 nmol/g dwt in ileal contents for the Control group to 40410 ± 3921 nmol/g dwt at Phy500 (P = 0.018) and to 10173 ± 2236 nmol/g dwt at Phy6000 (P<0.0001) (Table 4), with similar reduction compared to gizzard digesta response to phytase. Effects on InsP$_6$ levels were highly significant at both Phy500 and Phy6000 (for Phy500, P = 0.008; for Phy6000, P<0.0001). Here, InsP$_6$ was reduced from 51587 ± 3269 nmol/g dwt, for Control group, to 29190 ± 3803 nmol/g dwt and 1747 ± 385 nmol/g dwt, respectively.

Inositol levels of the gizzard and ileal contents were both impacted by inclusion of phytase in the diet (Tables 3 and 4), with increases in detectable free inositol with increasing phytase dose. Highly significant differences in inositol were measured at Phy6000 in both the gizzard and ileum (for the gizzard, P = 0.0002; for the ileum, P = 0.004). At Phy500, effects were significant in the gizzard but not in the ileum (P = 0.037 for gizzard and P = 0.102 for ileum). Inositol levels were measured at 348 ± 77 nmol/g dwt in the gizzard digesta for the Control group, 900 ± 147 nmol/g dwt at Phy500 and 2606 ± 284 nmol/g dwt at Phy6000. Inositol levels of the ileal contents were 1008 ± 297 nmol/g dwt for the Control group, compared with 2434 ± 654 nmol/g dwt for the Phy500 and 10870 ± 2233 nmol/g dwt for the Phy6000 groups. The observed greater effect on total inositol phosphates in the gizzard as opposed to the ileal digesta has been observed previously, and in these studies in which non-digestible markers have been used the effect may arise from the faster transit of soluble InsPs through the gizzard in comparison to the digestibility index marker and therefore subsequent apparent concentration in the terminal ileum.

### Profiles of tissue inositol phosphates

One objective of this study was to investigate the effect of the addition of dietary phytase on the inositol phosphate levels observed in poultry tissues. Previous studies have identified changes in plasma inositol levels in relation to changes in gizzard and ileal phytate hydrolysis [9, 20, 27], but have been unable to access tissue inositol phosphates by commonly used analytical methods. Consequently, the inositol phosphate profile of different tissues is undefined. Here, the use of TiO$_2$ beads to pre-concentrate inositol phosphates during extraction enabled

**Table 5. Inositol and inositol phosphate (InsP$_{2-6}$) levels (nmol/g wwt) in kidney of day 21 broilers[1,2].**

| Diet | Inositol | InsP$_2$ | InsP$_3$ | InsP$_4$ | InsP$_5$ | InsP$_6$ | Σ InsP |
|---|---|---|---|---|---|---|---|
| Control | 6430±480 | 1.4±0.2 | 3.3±1.0 | 27.0±5.4 | 40.7±6.0 | 15.4±2.4 | 87.9±12.5 |
| Phy500 | 6600±260 | 0.9±0.2 | 1.2±0.1 | 14.0±2.1 | 31.5±3.0 | 10.6±2.1 | 58.1±5.5 |
| Phy6000 | 7530±310 | 0.7±0.2 | 1.7±0.6 | 9.8±1.5 | 15.9±3.1 | 5.2±0.73 | 33.4±5.1 |
| | | | | Probabilities | | | |
| Control vs. Phy500 | 0.770 | 0.182 | 0.203 | 0.203 | 0.366 | 0.366 | 0.203 |
| Control vs. Phy6000 | 0.134 | 0.039 | 0.182 | 0.023 | 0.006 | 0.003 | 0.003 |
| Phy500 vs. Phy6000 | 0.130 | 0.611 | 0.570 | 0.317 | 0.013 | 0.120 | 0.019 |

Abbreviations: Σ InsP, total InsP$_2$ to InsP$_6$; InsP$_6$, inositol hexakisphosphate; InsP$_5$, inositol pentakisphosphate; InsP$_4$, inositol tetrakisphosphate; InsP$_3$, inositol trisphosphate; InsP$_2$, inositol bisphosphate.

[1]The control group was fed with a diet with 0.45% calculated available phosphate. Groups Phy500 and Phy6000 were fed with the control diet supplemented with 500 or 6,000 FTU of phytase per kilogram of feed, respectively.

[2]Data are given as group means ± SEM, n = 12, of 6 replicate pens with samples taken from 2 broilers per pen per treatment. Statistical analysis was performed by multiple T tests with correction for multiple comparisons using the Holm-Šídák method for inositol, inositol phosphate and Σ InsP data.

us in this study to measure inositol phosphate levels in combination with existing analytical methods.

Kidney tissue inositol phosphates (Table 5) show similar reduction as in digesta with increasing phytase. Ins(1,3,4,5,6)P$_5$ is the dominant inositol phosphate measured in these tissues followed by D-/and or L-Ins(3,4,5,6)P$_4$, with InsP$_5$ over 3-fold higher than InsP$_6$ in these samples (Fig 1B). Significant differences were measured between the Control and Phy6000 diets for: InsP$_4$ (P = 0.023), 27.0 ± 5.4 nmol/g wwt and 9.8 ± 1.5 nmol/g wwt, respectively; InsP$_5$ (P = 0.006), 40.7.0 ± 6.0 nmol/g wwt and 15.9 ± 3.1 nmol/g wwt, respectively; InsP$_6$ (P = 0.003), 15.4 ± 2.4 mnol/g wwt and 5.2 ± 0.73 nmol/g wwt, respectively. At Phy500, InsP$_4$, 14.0 ± 2.1 nmol/g wwt and InsP$_5$, 31.5 ± 3.0 nmol/g wwt were not statistically significantly different (P = 0.0592 and P = 0.2762, respectively).

Total inositol phosphate levels were reduced significantly (P = 0.003) in the kidney from a Control value of 87.9 ± 12.5 nmol/g wwt, to 33.4 ± 5.1 nmol/g wwt at Phy6000 (Table 5).

Ins(1,3,4,5,6)P$_5$ was the dominant inositol phosphate in liver (Table 6, Fig 1A), and although in this case InsP$_6$ was the next most abundant species, the identities of inositol phosphates were similar to kidney. Liver inositol phosphate levels showed a reduction in inositol phosphates from 36.5 ± 5.3 nmol/g wwt for the Control group to 24.3 ± 3.4 nmol/g wwt on addition of Phy500, though further reduction with increasing, Phy6000, was not observed, 30.8 ± 5.7 nmol/g wwt, and neither treatment was significantly different from Control group (for Phy500, P = 0.327; for Phy6000, P = 0.927).

Kidney inositol levels were not significantly affected by the addition of phytase to the control diet despite changes to inositol phosphate levels observed in the same sample (Table 5). Slight numerical increases in the inositol levels were measured in the kidney tissue between different dietary phytase doses, with 6430 ± 480 nmol/g wwt in the Control group; 6600 ± 260 nmol/g wwt with Phy500 and 7530 ± 310 nmol/g wwt with Phy6000. The differences were not significant (Control vs. Phy500, P = 0.182; Control vs. Phy6000, P = 0.133).

Similarly, sample inositol levels were increased in the liver tissue from 15920 ± 870 nmol/g wwt in the Control group to 16060 ± 550 nmol/g wwt at Phy500, and 18110 ± 990 nmol/g wwt at Phy6000, but again these differences were not significant (Control vs. Phy500, P = 0.964; Control vs. Phy6000, P = 0.507) (Table 6). In the study of Gonzalez-Uarquin et al. [31], a statistically significant increase in tissue inositol was observed in kidney of d 21 broilers at 1500

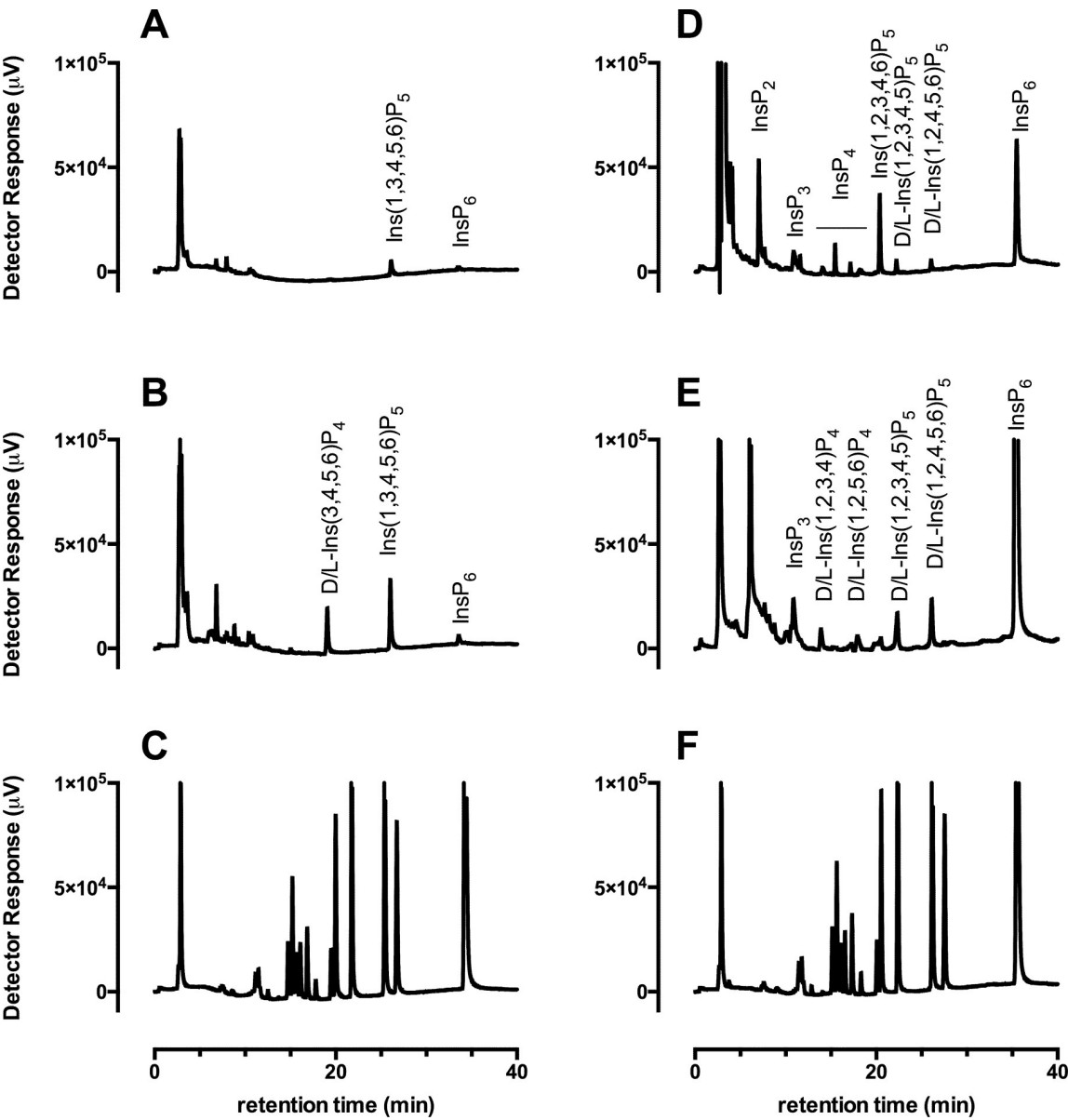

**Fig 1. Inositol phosphates in broiler digesta and tissue.** Extracts of (A), liver; (B), kidney; (D), gizzard contents; (E), ileal contents; from a single bird (A,B) or pooled from 2 birds (D,E) fed the Control diet were analysed by HPLC. (C) and (F), standards run beside the different sample sets from which A,B and D,E, respectively, were obtained. Inositol phosphate classes and individual isomers are identified.

FTU/kg, but not at 3000 FTU/kg, while liver levels of inositol did not differ between treatments.

## Inositol phosphates in avian tissues

The peaks identified in Fig 1B for kidney tissue samples are also the isomers present in liver tissue samples (Fig 1A). The identities of peaks in the set of standards (Fig 1C) have been described [20, 36, 39]. The isomers detected in tissues, D-and/or L-Ins(3,4,5,6)$P_4$ and Ins(1,3,4,5,6)$P_5$, differ from the known products of phytate degradation in Quantum Blue-

**Table 6. Inositol and inositol phosphate (InsP$_{2-6}$) levels (nmol/g wwt) in liver of day 21 broilers[1,2].**

| Diet | Inositol | InsP$_2$ | InsP$_3$ | InsP$_4$ | InsP$_5$ | InsP$_6$ | Σ InsP |
|---|---|---|---|---|---|---|---|
| Control | 15920±870 | n.d. | 2.2±0.1 | 1.8±0.3 | 23.6±4.3 | 8.9±1.4 | 36.5±5.3 |
| Phy500 | 16060±550 | n.d. | 2.3±0.4 | 1.3±0.3 | 15.1±2.2 | 5.5±1.1 | 24.3±3.4 |
| Phy6000 | 18110±990 | n.d. | 2.0±0.2 | 1.7±0.3 | 19.4±4.6 | 7.7±1.1 | 30.8±5.7 |
| | | | | Probabilities | | | |
| Control vs. Phy500 | 0.964 | - | 0.964 | 0.481 | 0.328 | 0.328 | 0.328 |
| Control vs. Phy6000 | 0.507 | - | 0.927 | 0.927 | 0.927 | 0.927 | 0.927 |
| Phy500 vs. Phy6000 | 0.408 | - | 0.808 | 0.808 | 0.808 | 0.601 | 0.808 |

Abbreviations: Σ InsP, total InsP$_2$ to InsP$_6$; InsP$_6$, inositol hexakisphosphate; InsP$_5$, inositol pentakisphosphate; InsP$_4$, inositol tetrakisphosphate; InsP$_3$, inositol trisphosphate; InsP$_2$, inositol bisphosphate.

[1]The control group was fed with a diet with 0.45% calculated available phosphate. Groups Phy500 and Phy6000 were fed with the control diet supplemented with 500 or 6,000 FTU of phytase per kilogram of feed, respectively.

supplemented diets, D-and/or L-Ins(1,2,3,4)P$_4$, D-and/or L-Ins(1,2,5,6)P$_4$, Ins(1,2,3,4,6)P$_5$, D-and/or L-Ins(1,2,3,4,5)P$_5$ and D-and/or L-Ins(1,2,4,5,6)P$_5$ [40] (Fig 1D and 1E). We comment that the diets were fed as mash without heat treatment, hence the presence of Ins(1,2,3,4,6)P$_5$, a known product of endogenous phytase in wheat-based diets not exposed to heat via pelleting.

## Discussion

The tissue inositol phosphates (Fig 1A and 1B) are similar to those identified in avian erythrocytes [20, 41, 42]. Thus, despite clear impact of dietary phytase on inositol phosphates in gizzard and ileal content, and on tissue inositol phosphates, the tissue isomers (of inositol phosphates) are phosphorylated in positions not expected from phytate degradation by Quantum Blue or endogenous feed phytase. Rather, they reflect the isomers expected from *de novo* inositol phosphate synthesis [42] and they match the isomers analysed in *Xenopus* and rat skeletal muscles [41]. It is widely accepted that products of phytate degradation retain phosphate in the 2-position. It is also widely accepted that the final step in inositol hexakisphosphate biosynthesis involves addition of phosphate to the 2-position [33]. Because the isomers observed in tissues lacked the 2-phosphate, they cannot arise simply by absorption from the gut, since those in the gut possess the 2-phosphate. While it cannot be excluded that potential selective InsP absorption and metabolism thereof contributes to the tissue profile observed, inositol phosphate transporters have not been described in animals. In contrast there are a plethora of studies describing expression profiles and biochemical properties of inositol and phosphate transporters in the gastro-intestinal tract of poultry. We conclude that the effect seen in different dietary conditions (0, Phy500, and Phy6000) does not arise from uptake of InsPs following gut phytate hydrolysis, but rather from tissue response to changing phosphate and/or inositol availability.

Kidney tissue inositol phosphate levels, as individual InsP$_4$, InsP$_5$ and InsP$_6$ isomer(s), as well as total inositol phosphate levels, decrease with increasing phytase dose. This suggests that the response arises from the influence of increasing free inorganic Pi and/or inositol in the gut and/or their tissue-specific influence on inositol phosphate biosynthetic gene expression. We are not aware of any studies of tissue response of inositol phosphates to circulatory inositol or phosphate, other than in blood [20]. Nonetheless, increases in circulatory inositol with dietary phytase are widely reported in poultry [9, 16, 17, 23–27].

In animals and humans, the kidney is the most important organ for maintenance of inositol concentration regulation in blood plasma [43], and studies in rat models suggest inositol catabolism occurs mainly in the kidney [44], though there is no research to suggest that the same models hold true for poultry tissues [27]. Likewise, there is no current consensus where the aggregate mass of inositol phosphate synthesis occurs or how circulatory inositol modifies tissue inositol synthesis and use. Nevertheless, as inositol and phosphate are the absorbable co-products of phytase action, it seems likely that the physiological response of poultry to phytase integrates the two. Response of poultry to phytase is most commonly interpreted in context of Ca and available phosphate with particular focus on amino acid digestibility and Ca: P [4, 26]. It is relevant, therefore, to put inositol provision in context of Ca and phosphate homeostasis. This study shows that in broilers kidney inositol phosphate levels are particularly responsive to dietary phytase, while liver levels are not.

Human and mammalian studies show that inter-organ signalling between the gut, kidney, Parathyroid gland (PTG) and bone, mediated by hormones, vitamin D, parathyroid hormone (PTH) and fibroblast growth factor 23 (FGF23), regulates phosphate homeostasis [45–48], with studies using mice models showing Na/Pi cotransporters in the small intestine and kidney are regulated by dietary phosphate [49]. Circulatory phosphate is controlled tightly and tissue phosphate is protected by phosphate resorption from bone under control of PTH: under conditions of hypocalcaemia the PTG increases PTH secretion, which decreases renal distal tubule calcium excretion and inhibits phosphate reabsorption in the proximal tubule [48]. In conditions of hyperphosphataemia, FGF23 produced by osteoblasts and osteocytes in response to PTH increases renal phosphate excretion by down-regulating the expression of NaPi-IIa and NaPiIIc cotransporters in the proximal tubules. Antibodies against FGF23 have been show to increase P retention in poultry [50].

In chickens, plasma vitamin D is predominantly 25-OH-$D_3$ with lower levels of 24,25-$(OH)_2$-$D_3$ [51]: the liver is the principal organ modifying cholecalciferol (vitamin $D_3$) by hydroxylation to 25-OH-$D_3$, the most active form in poultry [52], while further conversion (and inactivation) to 24,25-$(OH)_2$-$D_3$ by 24-OHase is mediated by an enzyme with highest expression in chicken kidney [53]. 1$\alpha$-OH-$D_3$ supplementation of broiler diets is purported to bypass the critical 1-hydroxylation of 25-OH-$D_3$ occurring in the kidney, by allowing 25-OHase of the liver to produce the vitamin D receptor- (VDR-) active ligand 1,25-$(OH)_2$-$D_3$ [54].

Vitamin D ($D_3$ predominantly) has been studied most extensively in poultry as additive to diet in context of tibial dyschondroplasia or egg production and there are relatively few reports of vitamin D or its metabolite levels in plasma or tissues of supplemented or non-supplemented birds [54, 55]. We are not aware of studies of effect of phytase on levels of vit D or its metabolites, though, we note synergistic effect of phytase and vit $D_3$ on growth performance, interpreted, in part, in context of effect of vit $D_3$ on intestinal phytase activity [56]. Earlier reports point to the efficacy of vit $D_3$ and its derivatives to increase phytate hydrolysis [57–59].

In laying hens, medullary bone is a highly labile source of calcium and phosphate for production of eggshell [60, 61]. The high calcium requirement of layers [12] will likely add further complication, beyond that elaborated here for broilers, to the intersection of the inositol and phosphate co-products (of phytase action) with phosphate homeostasis. Nevertheless, Greene et al. [15] have shown that the mRNA levels of various genes of inositol phosphate synthesis and turnover, inositol polyphosphate 1-phosphatase (INPP1), inositol hexakisphosphate kinase 1–2 (IP6K1-3), *myo*-inositol phosphate synthase (ISYNA) and multiple inositol polyphosphate phosphatase (MINPP1) are altered post-prandially in blood and feather of broilers by phytase. One implication is that tissue inositol phosphate synthesis/turnover is responsive to phytase, as recently shown for blood [20]. While some studies ascribe cellular function to

the IP6K1-3/InsP$_6$ product, 5-InsP$_7$, as regulator of plasma phosphate [62], evidence from this poultry trial suggests that physiologically the kidney is especially responsive, directly or indirectly, to phosphate and/or inositol from diet. The methods elaborated here, in their use of TiO$_2$ as a pre-concentration method for inositol phosphate analysis, will allow incisive testing of tissue inositol phosphate contribution to phosphate homeostasis by allowing simple experimental access to previously unobtainable inositol phosphate measurements.

## Supporting information

**S1 Data.**
(ZIP)

## Acknowledgments

We thank the Science Analytical Facilities, Faculty of Science, UEA for technical assistance.

Conflict of interests statement: AB Vista had no role in conducting the research, generating the data, interpreting the results or writing the manuscript.

## Author Contributions

**Conceptualization:** Emily Burton, Michael R. Bedford, Charles A. Brearley.

**Data curation:** Colleen Sprigg, Charles A. Brearley.

**Formal analysis:** Colleen Sprigg, Charles A. Brearley.

**Funding acquisition:** Michael R. Bedford, Charles A. Brearley.

**Investigation:** Colleen Sprigg, Hayley Whitfield, Emily Burton, Dawn Scholey.

**Methodology:** Colleen Sprigg, Hayley Whitfield, Emily Burton.

**Project administration:** Emily Burton, Dawn Scholey, Michael R. Bedford.

**Writing – original draft:** Colleen Sprigg.

**Writing – review & editing:** Emily Burton, Charles A. Brearley.

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
