## [Decision Letter · Decision Letter 0]

27 Apr 2022

PONE-D-22-04744Phytase dose-dependent response of kidney inositol phosphate levels in poultryPLOS ONE

Dear Dr. Brearley,

Thank you for submitting your manuscript to PLOS ONE. After careful consideration, we feel that it has merit but does not fully meet PLOS ONE’s publication criteria as it currently stands. Therefore, we invite you to submit a revised version of the manuscript that addresses the points raised during the review process.

I CONCUR WITH ALL THE COMMENTS RAISED BY THE THREE REVIEWERS. PLEASE ADDRESS THESE ISSUES IN THE REVISED MANUSCRIPT.

We look forward to receiving your revised manuscript.

Kind regards,

Juan J Loor

Academic Editor

PLOS ONE

Journal Requirements:

2. In your Methods section, please provide additional information on the animal research and ensure you have included details on : (1) methods of sacrifice (2) methods of anesthesia and/or analgesia, and (3) efforts to alleviate suffering.

3.In your Data Availability statement, you have not specified where the minimal data set underlying the results described in your manuscript can be found. PLOS defines a study's minimal data set as the underlying data used to reach the conclusions drawn in the manuscript and any additional data required to replicate the reported study findings in their entirety. All PLOS journals require that the minimal data set be made fully available. For more information about our data policy, please see http://journals.plos.org/plosone/s/data-availability.

Reviewers' comments:

Reviewer's Responses to Questions

**Comments to the Author**

1. Is the manuscript technically sound, and do the data support the conclusions?

Reviewer #1: Yes

Reviewer #2: Partly

Reviewer #3: Partly

2. Has the statistical analysis been performed appropriately and rigorously? 

Reviewer #1: Yes

Reviewer #2: No

Reviewer #3: Yes

3. Have the authors made all data underlying the findings in their manuscript fully available?

Reviewer #1: Yes

Reviewer #2: Yes

Reviewer #3: Yes

4. Is the manuscript presented in an intelligible fashion and written in standard English?

Reviewer #1: Yes

Reviewer #2: Yes

Reviewer #3: No

5. Review Comments to the Author

Reviewer #1: Sprigg et al Phytase dose-dependent response of kidney inositol phosphate levels in poultry

Phytase is one of the most used feed additives in animal nutrition, with benefits ranging from animal productivity, reduction of feed costs, reduced environmental impact. This work extends these benefits by study the impact at metabolic level generation of inositol phosphates in different parts of the gastro-intestinal tract, specifically exploring the role of phytase in release inositol phosphate in tissues. The authors provided clarity the data described here is part of a larger project to be published elsewhere.

L23, please delete one bracket; clarify what is meant by “larger 2x4 design.

L93-94; though not part of this study it would be informative to clarify why two levels of titanium were investigated

L100 (Table 1), are you certain the table displays analyzed or calculated nutrient? If analyzed describe how AME and NE were analyzed? Please note should be Crude fat not Fat. What does soy oil global mean? Is available P and Ca account for phytase contribution?

L107, please clarify declared crude protein

L114-115; previously indicated there two levels of titanium? Please clarify?

L117, please clarify the reason behind this does of inositol?

L118, italicize E. coli

L124/Table 2; please provide clear description of the diets, for example what is the difference between Control vs. control Ti, reader should be able to understand the table without referring to the text. Besides this why would you expect inositol concentrations to vary between Ti and non-Ti diets; variation is appreciable and the best way to present the data is mean±SD

L142, how were 6 replicates arrived at, was power analyses conducted to justify these number of replications?

L162-163, please add information on how the organs were processed prior to aliquoting 100 mg for analyses; ideally one would expect the whole organ homogenized and sub-sample taken

L166; should be mL not ml, please updated throughout the manuscript

L275; please explain this concept “The use of TiO2 as a pre-concentration method for inositol phosphates” also in relation of why data in Table 5 is not presented as diets with and without Ti

Reviewer #2: The manuscript is part of a study performed to investigate the impact of microbial phytase (0, 500 and 6000 FTU/kg) and inositol phosphate in the presence or absence of TiO2 on InsPx profiles in the gizzard and ileum digesta as well as in the kidney and liver of broilers. The major InsPx isomers found in the gizzard and ileum digesta were different from those in the liver and kidney, hence it was concluded that the InsPx in the kidney and liver was not directly absorbed from the digestive tract but synthesized de novo. This conclusion did not seem to be very well supported by the findings, because 1). It was unknown from this manuscript if InsPx isomers could be directly absorbed from the intestine or not, if so which transporters involved; 2). It could not be excluded that the intestine selectively absorbed some InsPx isomers and then directly deposit them in the kidney, resulting in the different InsPx isomers found in the kidney/live from those in the gizzard and ileum digesta. Next to that, experimental design and statistical analysis of the complete study were described, while only three treatments were selected for InsPx analysis. Please remove all treatments/tissues that were not used in this study. Lastly, there are many typo mistakes in the current manuscript, particularly in the tables.

Line 25: Only the kidney and liver tissues of three treatments were analyzed.

Line 28-33. The IP esters found in the tissue is different from those in the digesta, while the conclusion was not well supported. See comments above.

Line 109: what is Mb? Mg?

Line 114-122: Design of the whole study was described, while only three treatments were selected for analysis. Please only describe the treatments/tissues that were used in the analysis, to be consistent with the treatment groups shown in tables/results.

Line 124. InsP3 of the Phy6000 Ti group was much lower than other treatments, suggesting a lab mistake? By the way, this should be the measured inositol phosphate concentration in all treatment diets, not the basal diets.

Line 140 and 143. typo mistake, conflicting starting days, day 0 in line 140 and day 1 in line 143.

Line 158-162. Please only describe the tissues used for inositol phosphate analysis.

Please clarify how were the gizzard and ileum content collected?

Line 190-193: do not understand the eluting procedure.

Line 203-207. Please describe the statistical analysis used in this study, not the whole study because only three treatments were selected for analysis.

Line 228, Table 3: Why was the total InsP reduced with phytase inclusion in the gizzard? Could any of these InsPx or inositol be absorbed from the gizzard, if so, which transporter(s) involved?

Please add P values to the table.

Typo mistake, 0.188c

Line 236-238. data analysis description did not agree with the table 3-6.

Line 271-277. Already described in the materials and methods section.

Line 280. Table 5. several typo mistakes, 12.57a,,,,3.0ab

Please check the numbers and superscripts in the table

Line 298. typo mistake. Double ..

Line 295-296. significance was not shown with superscript letter in table 5.

Line 298-300. compared to the control group or the Phy6000 group?

Line 338, typo mistake, day 21.

Line 394-425. This part was actually more about whole body Ca and P homeostasis. It is a pity that a connection between Ca and P homeostasis and tissue InsPx metabolism was missing.

Maybe add a short paragraph to summarize the major findings and conclusions.

Reviewer #3: In this manuscript, the authors investigated the effect of phytase on the inositol/phosphate in gizzard, ileum, kidney, liver, and other tissues. The results are interesting, but the manuscript writing and data presentation are confusing and difficult to follow. The following are the specific concerns.

1. At line 93, what are the “2 x 4 factorial arrangement”? What are the 2 factors? And what are the 4 levels?

2. At lines 94-5, what are the “Two basal diets (Table 1)”? I only saw one diet.

3. At lines 114-122 and table 2, it is confusing, what were the dietary treatments? The dietary description made more confusing. Looks like 8 diets, based on Table 2. Please make a dietary table.

4. In the tables of inositol/phosphate, please use only one unit, either micro or nano mole. It is very confusing to switch the unit back and forth. In table 2, please summarize the total inositol/phosphate, just as in other tables. Are the values significant different between diets?

5. For the data in tables 3, 4, 5, and 6, it would be useful to compare the relative values of the digesta/tissue inositol/phosphate to the dietary levels, at least for the total values. The absolute values would be easily misinterpreted if without dietary values.

6. Please remove the subtitle in discussion sections and the discussion should be concise and focus on new findings and implications.

7. Please move Figure 1 to result section.

8. Please provide background information about what means for the inositol and InsP3-6 in the context of phytase? More phytase, more inositol in digesta and/or tissue?

6. PLOS authors have the option to publish the peer review history of their article (what does this mean?). If published, this will include your full peer review and any attached files.

Reviewer #1: **Yes: **Elijah G. Kiarie

Reviewer #2: No

Reviewer #3: No

---

## [Author Response · Author response to Decision Letter 0]

16 Jun 2022

To whom it may concern,

Re. Phytase dose-dependent response of kidney inositol phosphate levels in poultry 

Authors: Colleen Sprigg1, Hayley Whitfield1, Emily Burton2, Dawn Scholey2, Michael R. Bedford3, Charles A. Brearley1*

On behalf of all authors, I am pleased to return a revised manuscript that addresses the considered criticisms of our manuscript. We thank the reviewers and the editor for their inputs.

Our revisions address points raised by the Academic Editor:

1. We have edited our manuscript to meet PLOS ONE’s style requirements

2. Method of sacrifice, anaesthesia and efforts to alleviate suffering statement are provided in the text, new lines 128-149.

We additionally note:

The study was conducted feeding test materials previously investigated in a number of broiler studies and shown to cause no pain, suffering or lasting harm. The birds were not exposed to invasive experimental procedures during the study and received routine care and husbandry akin to a commercial broiler farm. No analgesia / anaesthetic was required, and no suffering occurred prior to death. The study was overseen by the site Named Animal Care and Welfare Officer, who is empowered to end the trial immediately if they believe any suffering occurs. Birds were health checked twice daily by a technical team not involved in the outcome of the study, and any sick or lame birds were euthanised immediately. A record table was used throughout the study to assure no deaths or illness were associated with any treatment and that mortality did not rise above the expected level for the age and strain of bird.

3. We have provided the raw data as excel files (Supporting Information) 

To address the points raised by Reviewer #1: 

L23 and L93-94: we have modified the description of the experimental design to emphasize the comparisons for which data is provided in this manuscript (new lines 22-23 and 92-95). Our reference to the use of titanium is limited to new lines 128-130.

L100: Table 1 has been corrected to include statement of calculated value, rather than measured; Crude fat and a clarifying footnote 3 states that Available phosphate does not account for the contribution of phytate P.

L107: the crude protein content is clarified as footnote 1

L114-115: is addressed in response to L93-94 above

L117: reference to inositol dose has been removed from the manuscript, because we restrict our analysis of effect of phytase dose. But, for our reviewer: in a typical diet there are approximately 2-2.5g of inositol per kg feed in the form of IP6 so if this were totally dephosphorylated through superdosing of phytase it would yield approximately 2-2.5g inositol

L118: corrected

L124: again (see L117, reference to treatments not discussed in this manuscript (inositol and titanium) have been removed. This accommodates the concerns of Reviewers #2 and #3.

L142: we provide below (not in the text) a statement of our power calculations. 

Data for mean gizzard and ileal inositol contents in responses to phytase additional with N and standard error reported by Walk et al., 2018 was used to conduct a power calculation indicating 6 replicates per treatment were sufficient to identify treatment differences at a power setting of 80% and a type 1 error rate of 5%.

https://www.sciencedirect.com/science/article/pii/S0032579119309800

Walk, C.L., Bedford, M.R. and Olukosi, OA. (2018) Effect of phytase on growth performance, phytate degradation and gene expression of myo-inositol transporters in the small intestine, liver and kidney of 21 day old broilers. Poultry Science 97(4) 1155-1162

L162-163: Tissue was sub-sampled from frozen organs, rather than extraction of the entire organ (new line 171). This approach was taken to allow trial development of methods appropriate for other classes of analyte not described in this study.

L166: corrected. 

 L275: while researchers have noted empirical difference in eg. Ileal digestibility with different digestibility markers, it has not been appreciated by the animal nutrition field that that cell biology/signalling community have adopted the use of titanium dioxide as a solid phase extraction media for inositol phosphates. Indeed, we had cited the relevant methods (here in new lines 178-184 [32]). We will return to the discussion of the suitability of TiO2 as a digestibility marker, for inositol phosphate analysis in digesta and tissues in a subsequent manuscript. But, to simplify this manuscript, for the reader, in line with the preferences of Reviewers #2 and #3, we have limited our discussion in the revised manuscript to 3 treatments only (control, 500 and 6000FTU/kg ). We hope this ‘compromise’ meets the needs of all our reviewers.

To address the points raised by Reviewer #2: 

1) and 2) we provide a more detailed explanation of inositol hexakisphosphate synthesis and inositol hexakisphosphate degradation by phytases (new lines 378-389) that argues at which point in metabolic sequences the diagnostic 2-phosphate is added or removed. The alternative, that inositol phosphates are taken up intact from the gut, cannot be discounted (and we say as much, new lines 382-383) but such a premise would necessarily be an invention since gut inositol phosphate transporters have not been described. Philosophically, the simplest explanation is the best – until transporters are identified.

2) We have removed all reference to treatments of the experimental design, for which data is not analysed.

We have corrected the assorted typos.

Line 25: we have corrected the sentence (new lines 25-26)

Lines 28-33: now addressed (see response to 1 and 2, above (new lines 378-389). We hope we have emphasized how isomeric identity (the 2-phosphate) is key to a full-understanding of the origins of isomers in tissues. A seminal reference [42] is cited (new line 378) to support the argument.

Line 109: corrected to Magnesium (new line 108)

Lines 114-122: we have amended the text to discuss the three treatments only (new lines 115-121).

Line 124: the titanium-containing diets have been removed from the analysis as requested (2) and Lines 114-122). We provide values of different IPs for the three diets in Table 2.

Lines 140 and 143, corrected

Lines 158-162: we have provided new details (new lines 153-169).

Lines 190-193: All details of our use of TiO2 beads to pre-concentrate and elute inositol phosphate (after Wilson et al [32]) are provided (new lines 178-192). In short, inositol phosphates bound to TiO2 under acid conditions can be eluted by change to alkaline pH.

Lines 203-207: the details of our statistical analyses (limited to Control, Phy500 and Phy6000 treatments) are provided (new lines 85-86 and 210-214). 

Line 228: we, and others, have observed that soluble IPs transit through the GI tract faster than the insoluble marker and comment on this (new lines 275-279). We note that inositol transporters have been most widely studied in the small intestine of poultry and that inositol phosphate transporters have not been described (see also 1) and 2), above).

P values have been added to the table.

Lines 236-238: now corrected

Lines 271-277: text removed

Line 280: the Table has been corrected

Line 298: typo corrected.

Lines 295-296: corrected

Lines 298-300: full details of the comparisons made are provided (as probabilities) in Table 3

Line 338: corrected

Lines 394-425: we agree with the sentiment but the literature is sparse. Consequently, we leave our discussion of the topic to (new lines 455-561). We remain convinced that our data and this manuscript will encourage cell biologists to look at the poultry nutrition literature and the animal nutritionists to look at the cell biology of inositol phosphates.

A short paragraph: we hope the concluding sentences (new lines 455-461) make the point.

To address the points raised by Reviewer #3: 

1.Line 93: we have now limited the statistical analysis to the three treatments Control, Phy500 and Phy6000 and removed confusing reference to other aspects of the feeding trial that are not reported here. This satisifies Reviewer #2 (Lines 114-122, 124). We retain mention of digestibility marker but emphasize it is not discussed further.

2. Lines 94-5: as above

3. Lines 114-122: now described (new lines 128-130 and made evident in Table 2)

4.: Table 2 is provided, we make no claim of significant difference

5.: Tables 3,4,5,6, now include summed inositol phosphates, all reported in the same unit

6.: Subtitles have been removed, the discussion remains largely the same.

7.: Figure 1 has been moved

8.: Table 4 and Table 5 and the associated description (new lines 266-279) makes the point that as phytase dose increases and inositol phosphates are reduced in the gizzard and ileal digesta so the gizzard and ileal inositol increases. This is consistent with the literature (new lines 285 and 398, references [9,20,27] and [9,16,17,23-27], respectively. 

We thank our reviewers for their helpful comments on our manuscript.

Yours faithfully

Charles Brearley

---

## [Decision Letter · Decision Letter 1]

22 Aug 2022

PONE-D-22-04744R1Phytase dose-dependent response of kidney inositol phosphate levels in poultryPLOS ONE

Dear Dr. Brearley,

Thank you for submitting your manuscript to PLOS ONE. After careful consideration, we feel that it has merit but does not fully meet PLOS ONE’s publication criteria as it currently stands. Therefore, we invite you to submit a revised version of the manuscript that addresses the points raised during the review process.

PLEASE ADDRESS THE FINAL COMMENTS. I WILL MAKE A FINAL DECISION AFTERWARDS.

We look forward to receiving your revised manuscript.

Kind regards,

Juan J Loor

Academic Editor

PLOS ONE

Reviewers' comments:

Reviewer's Responses to Questions

**Comments to the Author**

1. If the authors have adequately addressed your comments raised in a previous round of review and you feel that this manuscript is now acceptable for publication, you may indicate that here to bypass the “Comments to the Author” section, enter your conflict of interest statement in the “Confidential to Editor” section, and submit your "Accept" recommendation.

Reviewer #2: All comments have been addressed

Reviewer #4: All comments have been addressed

Reviewer #5: All comments have been addressed

2. Is the manuscript technically sound, and do the data support the conclusions?

Reviewer #2: Yes

Reviewer #4: Partly

Reviewer #5: Yes

3. Has the statistical analysis been performed appropriately and rigorously? 

Reviewer #2: Yes

Reviewer #4: Yes

Reviewer #5: Yes

4. Have the authors made all data underlying the findings in their manuscript fully available?

Reviewer #2: Yes

Reviewer #4: Yes

Reviewer #5: Yes

5. Is the manuscript presented in an intelligible fashion and written in standard English?

Reviewer #2: Yes

Reviewer #4: Yes

Reviewer #5: Yes

6. Review Comments to the Author

Reviewer #2: (No Response)

Reviewer #4: 1 Abstract was not associated with the results, especially L27-29. There was no linear or quadratic analysis in the statistical analysis part. At the same time, no P value of liner relationship was shown in results table.

2 How many birds were used in this study? The number was not correct.

3 In introduction, the hypothesis of this study was not clear. From L67 to L80, these parts had no strong relationship with the object of the study.

4 L85 what kind of design? 2*4 or 3 treatments? So many statements of experiment design were not consistent.

5 there was no direct results of plasma vitamin D content in this study. L475-L482 the discussion of vitamin D metabolism in chicken was not suitable with less logic.

6 L494-L503 which results was discussed in this paragraph? Many references mentioned mRNA expressions during calcium and phosphate metabolism, what’s the relationship with the results?

7 in conclusion part, this study suggested that TiO2 was a better preconcentration of method for inositol phosphate analysis, but less discussion was mentioned. Thus, the conclusion of this study need rewrite.

Reviewer #5: I believe the authors have addressed all the comments adequately, and I have no further comments as a reviewer reviewed the respsonses of the authors to the other reviewers' comments.

7. PLOS authors have the option to publish the peer review history of their article (what does this mean?). If published, this will include your full peer review and any attached files.

Reviewer #2: No

Reviewer #4: No

Reviewer #5: No

---

## [Author Response · Author response to Decision Letter 1]

26 Aug 2022

PONE-D-22-04744R1

Phytase dose-dependent response of kidney inositol phosphate levels in poultry

PLOS ONE

To The Editor,

We thank our reviewers for their consideration of our manuscript and the opportunity afforded to improve the manuscript further. We hereby respond to the comments of Reviewer #4: 

1. Abstract was not associated with the results, especially L27-29. 

We think the reviewer has not appreciated that L27-29 make reference to the results described in lines 374-382: paragraph beginning, ‘ The peaks identified in Fig 1B…’ and further elaborated in the Discussion, paragraphs 1 and 2, lines 386-411.

There was no linear or quadratic analysis in the statistical analysis part. At the same time, no P value of liner relationship was shown in results table. 

The reviewer is correct, testing makes clear that the relationship is non-linear, we have therefore removed references to linearity – changing the term to one ‘progressive’- that does not need statistical validation.

2. How many birds were used in this study? The number was not correct. The trial as a whole was a 2 x 4 factorial design, of 8 treatments, designed to test 3 separate hypotheses. Data presented in this study speaks to only 3 of these treatment groups, with 60 birds per treatment in 6 pens of 10 birds per pen, totalling 180 birds. 

This has been further clarified in lines 146-149.

3. In introduction, the hypothesis of this study was not clear. From L67 to L80, these parts had no strong relationship with the object of the study. 

We have clarified this by addition of a sentence, lines 81-83, beginning, ‘It was hypothesised that ….’.

4. L85 what kind of design? 2*4 or 3 treatments? So many statements of experiment design were not consistent. 

This has been further clarified in lines 146 to 149

5. there was no direct results of plasma vitamin D content in this study. L475-L482 the discussion of vitamin D metabolism in chicken was not suitable with less logic. 

We choose to retain the Discussion unaltered and make the point that the Discussion puts our results in context of calcium and phosphate homeostasis and inter-organ axis for its control. Put simply, we are the first to describe effect of phytase on the inositol phosphates of any tissue, let alone kidney. That the kidney is central to phosphate and calcium homeostasis deserves the discussion of how the kidney is involved in calcium and phosphate homeostasis. We do believe that the assorted readers of this manuscript will hold a range of perspectives that make our Discussion relevant to their interests.

6. L494-L503 which results was discussed in this paragraph? Many references mentioned mRNA expressions during calcium and phosphate metabolism 

We think the reviewer has not appreciated that Discussion, paragraph 7 specifies a study (Greene et al. (15)) that describes effects of phytase on inositol phosphate synthesis genes. These genes should influence inositol phosphate levels and our study measures inositol phosphates in tissues. 

what’s the relationship with the results? 

Discussion, paragraphs 2 and 3, explicitly places the observed results for kidney inositol phosphate changes in the context of phosphate homeostasis under the prevailing sufficient phosphate and calcium provision conditions of this study. 

7. in conclusion part, this study suggested that TiO2 was a better preconcentration of method for inositol phosphate analysis, but less discussion was mentioned. Thus, the conclusion of this study need rewrite. 

We have expanded the last sentence of the last paragraph of Discussion, but retain the focus on the results that this method reveals (rather than the method itself, which we properly report as having been published previously by Wilson et al. 2015, but which has not been used for tissue/organ purpose). 

Finally, we reiterate the comments of Reviewer #5: I believe the authors have addressed all the comments adequately, and I have no further comments as a reviewer reviewed the respsonses of the authors to the other reviewers' comments.

Yours faithfully

Charles Brearley

---

## [Editor Report · Decision Letter 2]

22 Sep 2022

Phytase dose-dependent response of kidney inositol phosphate levels in poultry

PONE-D-22-04744R2

Dear Dr. Brearley,

We’re pleased to inform you that your manuscript has been judged scientifically suitable for publication and will be formally accepted for publication once it meets all outstanding technical requirements.

Kind regards,

Juan J Loor

Academic Editor

PLOS ONE
---

## [Editor Report · Acceptance letter]

10 Oct 2022

PONE-D-22-04744R2 

Phytase dose-dependent response of kidney inositol phosphate levels in poultry 

Dear Dr. Brearley:

I'm pleased to inform you that your manuscript has been deemed suitable for publication in PLOS ONE. Congratulations! Your manuscript is now with our production department. 

Kind regards, 

on behalf of

Dr. Juan J Loor 

Academic Editor

PLOS ONE